# Long-Term Degradation Evaluation of the Mismatch of Sensitive Capacitance in MEMS Accelerometers

**DOI:** 10.3390/mi14010190

**Published:** 2023-01-12

**Authors:** Xinlong Huang, Xianshan Dong, Guizhen Du, Youwang Hu

**Affiliations:** 1College of Mechanical and Electrical Engineering, Central South University, Changsha 410083, China; 2Science and Technology on Reliability Physics and Application Technology of Electronic Component Laboratory, China Electronic Product Reliability and Environmental Testing Research Institute, Guangzhou 511370, China; 3Institute of Advanced Wear & Corrosion Resistance and Functional Materials, Jinan University, Guangzhou 510632, China

**Keywords:** MEMS accelerometer, mismatch of sensitive capacitance, long-term reliability

## Abstract

During long-term use, MEMS accelerometers will experience degradation, such as bias and scale factor changes. Bias of MEMS capacitive accelerometers usually comes from the mismatch of parasitic capacitance and sensitive capacitance. This paper focuses on the mismatch of sensitive capacitance and analyzes the mechanism of long-term degradation of MEMS accelerometers. Firstly, the effect of sensitive capacitance mismatch on the performance of a MEMS accelerometer was investigated. Secondly, a method of measuring the mismatch of sensitive capacitance was proposed, and the validation experiment shows that the accuracy of this measurement can be less than $1.10\times 10^{-5}$ of the sensitive capacitance. For the samples in this experiment, the measurement error of this method can be less than 0.36 fF. Finally, a high-temperature acceleration experiment was performed. The mismatch of the sensitive capacitance during the experiment was monitored based on the proposed method, and the experimental results are analyzed. The experimental result demonstrates that the mismatch of sensitive capacitance varies linearly with time. The change rates of sensitive capacitance mismatch for the two samples are $2.95\times 10^{-7}\;C_{0}/h$ and $2.66\times 10^{-7}\;C_{0}/h$ in the high-temperature acceleration experiment at 145 °C, respectively. The change in sensitive capacitance mismatch seems small, but it is not to be ignored during long-term use. The rate of change is similar for the same batch of samples. This could imply that the adverse effects due to the mismatch of sensitive capacitance changes can be reduced by compensating for this variation.

## 1. Introduction

A MEMS accelerometer is an inertial navigation device which combines microelectronics and micro-machining technology. It is widely applied to various fields, such as aerospace [1,2], autonomous driving [3], oil exploration [4], and medical devices [5,6]. A capacitive accelerometer is a typical MEMS accelerometer because of the outstanding performance of small volume, low power consumption, high resolution, and wide dynamic range, etc. The sensitive structure of a capacitive accelerometer can be considered as a pair of differential capacitors [7,8]. Due to process error, residual stress, and environmental stress, the values of the two sensitive capacitors cannot be perfectly equal [9,10]. It can cause mismatch of sensitive capacitance in a micro-accelerometer, which would deteriorate the bias and linearity [11,12].

In military and automotive applications, MEMS accelerometers commonly require a lifetime of more than ten years. Therefore, the long-term stability of MEMS accelerometers should not be neglected. Onen A.S. et al. designed a temperature cycling experiment to simulate the aging process of the accelerometer of the IMU for three years [13]. The experimental result demonstrated that the bias would shift due to the thermal stress in the temperature cycling. Liu Y. monitored the bias instability of MEMS gyroscopes during high-temperature acceleration experiments [14]. They found that the bias instability of the MEMS gyroscope increased approximately linearly with time. Luczak S. et al. considered that the aging phenomenon of accelerometers could be related to four specific components: mechanical components, electric components, electronic circuits, and external electronic components [15]. However, they did not carry out a separate experiment on specific components.

In order to improve the comprehensive performance of accelerometers, many scholars have paid attention to the effect of capacitive mismatch of accelerometers and carried out some experiments. Ding Z. measured the mismatch of sensitive capacitance for accelerometers directly through the MS3110 chip and analyzed the effect of sensitive capacitance mismatch on bias [16]. However, they ignored the parasitic capacitance between the electrodes in the readout circuit. Chen D. et al. proposed a method to identify and calibrate parasitic mismatches based on a digital, closed-loop EM-Δ (electromechanical sigma delta) accelerometer system [17]. Consequently, their method is only applicable to closed-loop configurations. In addition, current research has focused more on the parasitic capacitance mismatch of capacitive MEMS accelerometers. The study on the sensitive capacitance mismatch of MEMS accelerometers is scarce and needs to be further explored.

In this paper, a method is proposed for measuring the mismatch of sensitive capacitance in a MEMS accelerometer. This method can avoid the interference of parasitic capacitance in the readout circuit, and the effectiveness of the proposed method is verified by experiment. The validation experiment shows that the measurement accuracy of the proposed method is better than 0.36 fF. It has some practical value, although it can only be used in an open-loop configuration. In addition, a high-temperature acceleration experiment was carried out based on the pendulum micro-accelerometer. The variation of the sensitive capacitance mismatch was analyzed. This paper is helpful for designers who need to analyze the long-term stability of MEMS accelerometers in terms of sensitive capacitance mismatch.

## 2. Theory

### 2.1. The Mismatch of Sensitive Capacitance

There are many types of capacitive accelerometers, such as comb accelerometers, pendulous accelerometers, and sandwich accelerometers [18,19]. The pendulum accelerometer has a larger overlapping area between the electrode plates, so it has a higher sensitivity compared to other capacitive accelerometers. Figure 1 shows the simplified physical model of the pendulum accelerometer. There are some blind holes on one side of the proof mass, which makes the mass center deviate from the torsional beam. When an input acceleration occurs in the sensitive direction, the proof mass will deflect, which causes a change in the sensitive capacitance of *C*_1_ and *C*_2_. The input acceleration can be obtained by measuring the change in the sensitive capacitances.

In practical applications, the deflection angle of the mass block is very small, and the capacitance error caused by the tilt of the electrode plate can be ignored. In order to facilitate the calculation, we simplify the sensitive structure of the pendulum accelerometer as shown in Figure 2.

Ideally, the distance between the proof mass and the $C_{1}$ electrode plate is the same as the $C_{2}$ electrode plate, and the distance is $d_{0}$. However, because of process error and residual stress, there is a distance $x_{0}$ between the proof mass and the ideal position. After ignoring the higher order terms, the difference between the two sensitive capacitors can be expressed as [20]:(1)$$∆C=\frac{\varepsilon_{r}\varepsilon_{0}S}{\left(d_{0}-x\right)}-\frac{\varepsilon_{r}\varepsilon_{0}S}{\left(d_{0}+x\right)}\approx K_{1}x_{1}+K_{2}x_{1}{}^{3}+K_{3}x_{0}+K_{4}x_{0}{}^{3}$$
where $x=x_{0}+x_{1}$ is the total offset, and $x_{1}$ is the component caused by the input acceleration. $\varepsilon_{r}$ and $\varepsilon_{0}$ are the relative permittivity and absolute permittivity, respectively. *S* is the overlapped area of sensitive capacitance. $K_{1}=K_{3}=2\varepsilon_{r}\varepsilon_{0}S/d_{0}{}^{2}$ is the scale factor of displacement conversion capacitance. $K_{2}=K_{4}=2\varepsilon_{r}\varepsilon_{0}S/d_{0}{}^{4}$ is the third-order nonlinear coefficient.

The differential capacitance can be expressed as the sum of four terms. The first term is linearly dependent on the input acceleration, which is the information we expect. The second term is the third-order nonlinear error caused by the input acceleration, which can be eliminated by the compensation circuit. The sum of the third and fourth items is the mismatch of the sensitive capacitance, which comes from the initial position error of the proof mass. Obviously, the mismatch of the sensitive capacitance would affect the bias of the MEMS accelerometer.

### 2.2. Method for Measuring the Mismatch of Sensitive Capacitance

In order to study the change in sensitive mismatch capacitance of MEMS accelerometers in long-term use, it is necessary to accurately measure the sensitive mismatch capacitance. Since the mismatch of the sensitive capacitance is very small, it is difficult to measure it accurately by existing measurement methods. In this paper, a method for measuring sensitive mismatch capacitance applicable to MEMS capacitive accelerometers is presented.

The principle of this method is to use the $C_{1}$ electrode plate and $C_{2}$ electrode plate to apply electrostatic force to the proof mass, so that the proof mass remains in the free state without the input acceleration. If the voltages of the $C_{1}$ electrode plate and $C_{2}$ electrode plate are not large, the average distance between electrode plates due to electrostatic forces varies very little. In this paper, the average distance between electrode plates is assumed to be constant. Based on the voltages of the $C_{1}$ electrode plate and the $C_{2}$ electrode plate, the offset distance of the proof mass from the ideal position can be obtained, and then the mismatch of the sensitive capacitance can be deduced.

In order to adjust the position of the proof mass, the open-loop measuring circuit is used to obtain the output signal of the micro-accelerometer. The simplified electrical model of the measurement circuit is shown in Figure 3. $V_{T}$ and $V_{B}$ are the voltage of the $C_{1}$ electrode plate and $C_{2}$ electrode, respectively. They are DC voltages, providing electrostatic force to the proof mass. $V_{d}$ is an AC square-wave signal for modulation and demodulation of the measurement circuit. The core component of the measurement circuit is the charge amplifier, which can convert the differential value between the two sensitive capacitors into a voltage signal. In addition, the printed circuit board inevitably has many parasitic capacitances due to the layout of metal alignment.

Obviously, the measuring circuit contains many parasitic capacitors that would affect the output signal of the MEMS accelerometer. $C_{p1}$ is the parasitic capacitance between the input of the charge amplifier and the $C_{1}$ electrode plate, and $C_{p2}$ is the parasitic capacitance between the input of the charge amplifier and the $C_{2}$ electrode plate. $C_{p3}$ is the parasitic capacitance between the proof mass and the ground. It should be noted that the mismatch of the sensitive capacitance cannot be measured directly, because these parasitic capacitances would also be measured, resulting in measurement error.

The output signal of the measuring circuit is directly related to the input charge of the charge amplifier. The input charge is sourced from the difference in charges during charging and discharging of $V_{d}$ to the two sensitive capacitors, and it can be calculated by the following Equation (2):(2)$$Q=\left(C_{1}+C_{p1}–C_{2}–C_{p2}\right)\cdot V_{d}$$

$C_{p1}$ and $C_{p2}$ are determined by the structure of the printed circuit board and do not change. The values of $C_{1}$ and $C_{2}$ would be affected by $V_{T}$, $V_{B}$, and the input acceleration. Therefore, we can keep the input acceleration at 0 g and adjust $V_{T}$ and $V_{B}$ to change the output of the capacitive accelerometer.

Firstly, $V_{T}$ and $V_{B}$ are set to 0 V, and the bias without electrostatic force is recorded as $U_{0}$. Then, $V_{T}$ and $V_{B}$ are adjusted to introduce two electrostatic forces. When the output of the capacitive accelerometer is again equal to $U_{0}$, the combined force of these two electrostatic forces is zero. It can be expressed as:(3)$$F_{e}=\frac{\varepsilon_{r}\varepsilon_{0}S\times V_{T}{}^{2}}{2d_{1}{}^{2}}-\frac{\varepsilon_{r}\varepsilon_{0}S\times V_{B}{}^{2}}{2d_{2}{}^{2}}=0$$
where $d_{1}$ is the distance between the proof mass and the $C_{1}$ electrode plate; $d_{2}$ is the distance between the proof mass and the $C_{2}$ electrode plate. The ratio of $d_{1}$ and $d_{2}$ can be expressed as:(4)$$\alpha =\frac{d_{1}}{d_{2}}=\frac{V_{T}}{V_{B}}$$

Therefore, $x_{0}$ can be calculated by the following equation:(5)$$x_{0}=\frac{\left(1-\alpha \right)}{\left(\alpha +1\right)}∆d_{0}$$

Then, the mismatch of the sensitive capacitance in MEMS accelerometer could be obtained:(6)$$∆C=C_{T}-C_{B}=\frac{\varepsilon_{r}\varepsilon_{0}S}{\left(d_{0}-x_{0}\right)}-\frac{\varepsilon_{r}\varepsilon_{0}S}{\left(d_{0}+x_{0}\right)}=\left(\frac{1-\alpha^{2}}{2\alpha }\right)C_{0}$$
where $C_{0}$ is the design value of the sensitive capacitor, $C_{0}=\frac{\varepsilon_{r}\varepsilon_{0}S}{d_{0}}$, which can be obtained from the product description manual of the MEMS accelerometer.

## 3. Experiment

### 3.1. Experiment Program

In this paper, the experiment was divided into three parts. Firstly, the validation experiment was designed to ensure that the method for measuring the mismatch of sensitive capacitance was valid. Secondly, a temperature tolerance experiment was carried out after referring to the Arrhenius acceleration model. It would determine the experimental temperature in the acceleration experiment. Finally, a high-temperature acceleration experiment was performed, and the mismatch of sensitive capacitance was analyzed based on the experimental data.

A batch of pendulum accelerometers with two sensitive capacitances of 11 pF were used as experimental samples. Two experimental samples were arranged in the validation experiment, and the sensitive capacitance mismatch of each sample was measured independently ten times based on the proposed method. A sample was used in a temperature tolerance experiment. It was successively treated at 25 °C, 85 °C, 105 °C, 125 °C, and 145 °C for 1 h, and its scale factor was measured at each state point. Two samples were applied in a high-temperature acceleration experiment, and the sensitive capacitance mismatches of these samples were monitored during the experiment. The experimental process is shown in Figure 4.

### 3.2. Validation Experiment

In order to verify the availability of the proposed method, a validation experiment was designed. In this experiment, a batch of capacitive accelerometers was used as an experimental sample, which was produced from 214 Institute of China North Industries Group. The sensitive structure of the experimental sample was a pendulum structure with a single anchor, similar to that shown in Figure 1. The characteristic parameters of the experimental sample are shown in Table 1.

Experimental samples were welded on the adapter board. The adapter board needs to be inserted into the measurement board and the measurement board is fixed on the turntable when measuring the mismatch of sensitive capacitance. The experimental sample and the measurement board are shown in Figure 5.

The measuring system consisted of the turntable with temperature control, multi-channel power supply, and digital multimeter 34,470 A, as shown in Figure 6. The turntable could adjust the sensitive direction of the experimental samples, so that their input acceleration was zero.

The measurement procedure for the sensitive capacitance mismatch of sample #1 was as follows:

(1) After the turntable was powered on and reset, the experimental sample was installed and fixed on the horizontal axis of the turntable; the angle of the turntable was adjusted so that the input acceleration of the accelerometer was zero;

(2) The chamber temperature of the turntable was set at 25 °C. The measuring board was powered on and the voltage of $C_{1}$ electrode plate and $C_{2}$ electrode plate was set to 0V, then 10 min elapsed for the sample to reach thermal stability;

(3) The output of the sample was recorded for 30 s by digital multimeter 34470 A, and its average was recorded as $U_{0}$;

(4) The voltages of the $C_{1}$ electrode plate and $C_{2}$ electrode plate were adjusted to make the output of the sample equal to $U_{0}$ again. However, limited by the resolution of the power supply output, it was difficult to make the output of the sample exactly equal to $U_{0}$ by adjusting the voltage of $C_{1}$ electrode plate and $C_{2}$ electrode plate. Therefore, we had to set the voltage of the $C_{2}$ electrode plate to a constant value and then adjust the voltage of the $C_{1}$ electrode plate so that the output of the sample was as close as possible to $U_{0}$.

In this step, the voltage of the $C_{2}$ electrode plate was set to 5 V. Table 2 shows the output of sample #1 at different electrode plate voltages, where $V_{T}$, $V_{B}$ was the voltage of $C_{1}$ electrode plate and $C_{2}$ electrode plate, respectively. $U_{0\mathrm{ut}}$ was the output of the sample.

It can be seen that $V_{B}$ is actually 5.00035 V. The $V_{T}$ that brings the proof mass into electrostatic balance can be calculated by the interpolation method:(7)$$V_{T}=5.00870+\frac{\left(0.17576-0.17971\right)\times \left(5.01192-5.00870\right)}{0.17360-0.17971}=5.01078\;V.$$Therefore, the ratio of the distance between the proof mass and the $C_{1}$ electrode plate and $C_{2}$ electrode plate can be expressed as:(8)$$\alpha =\frac{V_{T}}{V_{B}}=1.00209$$

According to Equation (6), the sensitive capacitance mismatch for sample #1 can be obtained:(9)$$∆C=\left(\frac{1-\alpha^{2}}{2\alpha }\right)C_{0}=-2.09\times 10^{-3}C_{0}=-22.97\;fF$$

In order to evaluate the measurement accuracy of the measurement method, the sensitive capacitance mismatch of each sample was measured 10 times. The electrode plate voltages of the sample at electrostatic force balance are recorded in Table 3. Their standard deviations (STD) of $V_{T}/V_{B}$ were calculated based on the 10 measurements.

Their mismatches of sensitive capacitance can be obtained according to Equations (4)–(6), as shown in Figure 7. It is worth noting that there were positive and negative mismatch values for the sensitive capacitance, which can indicate the direction of deflection of the proof mass. For example, a positive mismatch value of sensitive capacitance indicates that the proof mass is biased towards the $C_{1}$ electrode plate.

The maximum value of standard deviation was 1.10 × 10^−5^ C_0_, and the sensitive capacitance of this batch of samples was 11 pF. Therefore, the measurement error of the sensitive capacitance mismatch of these samples can be less than 0.36 fF according to the 3σ rule. It shows that the method has a good accuracy for measuring the mismatch of sensitive capacitance.

### 3.3. High-Temperature Acceleration Experiment

The principle of this accelerated experiment was to use the degradation rate in high environmental stress to deduce the degradation rate in normal environmental stress. The Arrhenius model is the most typical and widely used acceleration model, and its expression is shown in Equation (7) [21]:(10)$$\frac{\partial M}{\partial t}=A_{0}\cdot \exp\left(-\frac{∆E}{kT}\right),$$
where *M* is the degradation of the degradation parameter of the experimental sample, *k* is the Boltzmann constant, and *k* = 8.617 × 10^−5^ eV/K; ∆*E* is the activation energy of the experimental sample in the failure mechanism; $\frac{\partial M}{\partial t}$ is the degradation rate of the experimental sample at temperature *T*. Thus, the higher temperature can increase the efficiency of the accelerated experiment. In order to explore the appropriate experimental temperature of the batch of samples, a temperature tolerance experiment was designed. An experimental sample was successively treated at 85 °C, 105 °C, 125 °C, and 145 °C for one hour. A performance test was performed after the sample had returned to room temperature. The process of the temperature tolerance experiment is shown in Figure 8.

The scale factor of the experimental sample was obtained in the performance test and recorded in Table 4. As we can see, the change ratio of the scale factor is 0.07%, which illustrates that the performance of the sample remains unchanged in the temperature range from 25 °C to 145 °C, and the degradation mechanism does not change significantly. In order to reduce the cost of time in high-temperature accelerated experiments, the ambient temperature of the acceleration experiment is 145 °C.

In the high-temperature acceleration experiment, two samples were treated at 145 °C for 1500 h. The high-temperature accelerated test chamber as shown in Figure 9. Their mismatches of sensitive capacitance were measured before the high-temperature acceleration experiment, and this parameter was monitored during the high-temperature accelerated experiment to analyze the variation trend during long-term degradation.

## 4. Results and Discussion

In order to investigate the change process of the sensitive capacitance in the high-temperature acceleration experiment, the sensitive capacitance mismatch of each experimental sample was measured at 11 different time points. The sensitive capacitance mismatch of each sample at a time point was measured twice based on the proposed method, and their average values were calculated and recorded in Table 5.

The sensitive capacitance mismatch of the pendulum accelerometer is mainly related to the deflection of the proof mass. Process error, the degradation of torsional beam stiffness, and residual stress can cause the deflection of proof mass. The process error is a characteristic of the sensitive structure of each sample, which would not change in the high-temperature accelerated experiment. The torsion beam stiffness would gradually reduce during long-term use, but the change is small enough to be negligible. In the high-temperature acceleration experiment, the decrease in residual stress is the main reason for the variation of sensitive capacitance mismatch.

Figure 10 displays the variation of the sensitive capacitance of the experimental samples. During the high-temperature acceleration experiment of 1500 h, the sensitive capacitance mismatch of the two samples changed, and their changes were $+4.42\times 10^{-4}\;C_{0}$, $+3.91\times 10^{-4}\;C_{0}$, respectively. It should be noted that the sensitive capacitance mismatch is a vector, which can indicate the initial position and the trend of the proof mass. For example, the sensitive capacitance is a positive number, which means that the sensitive capacitance *C*_1_ is bigger than *C*_2_. In other words, the mass block is closer to the *C*_1_ electrode plate in this case. The schematic diagram of the position of the proof mass in the high temperature acceleration experiment is shown in Figure 11. It is obvious that the proof masses of the two samples gradually approach the $C_{1}$ electrode plate during the acceleration experiment. The reason for this phenomenon could be the asymmetric mass of the sensitive structure, which introduces an eccentric moment during the manufacturing process and is eventually embedded in the torsional beam; this eccentric moment gradually disappears during the high-temperature accelerated experiment.

In addition, the experimental results show that the change of sensitive capacitance mismatch is linear with time. Their rates of change in the acceleration experiments are $2.95\times 10^{-7}\;C_{0}/h$ and $2.66\times 10^{-7}\;C_{0}/h$, respectively. This trend may seem small, but it should not be ignored in long-term use. The difference between them is only 9.83%. It means that the degradation rate of a batch of samples in the same working environment is similar. Furthermore, we can predict the sensitive capacitance mismatch of a batch of MEMS accelerometers by measuring the change rate of sensitive capacitance mismatch for some samples. A programmable chip can also be added to the compensation circuit of the MEMS accelerometer for processing the original output of the accelerometer. The programmable chip can remove the component coming from the change in sensitive capacitor mismatch. Consequently, the adverse effect due to the change in the sensitive capacitor mismatch may be reduced by a compensation circuit.

## 5. Conclusions

In this paper, we propose a method to measure the sensitive capacitance mismatch of a capacitive accelerometer. The method was verified by experiment. In order to study the variation trend of the sensitive capacitance mismatch of the MEMS accelerometer during long-term use, a high-temperature acceleration experiment was designed. The measurement data were obtained, and the measurement results were analyzed. The conclusions in this paper can be summarized as follows:

(1) The measurement method of sensitive capacitance mismatch based on electrostatic balance is available and has good accuracy. The validation experiment shows that the measurement error of this method can be less than 1.10 × 10^−5^ *C*_0_;

(2) In the high-temperature acceleration experiment, the mismatch value of sensitive capacitance of the MEMS accelerometer would change, and it varies linearly with time;

(3) For MEMS accelerometers in the same batch, the change rate of the sensitive capacitance mismatch is similar. According to this rate of change, a compensation circuit could be designed and used to reduce the adverse effect caused by the variation of sensitive capacitance mismatch.

## Figures and Tables

**Figure 1 micromachines-14-00190-f001:**
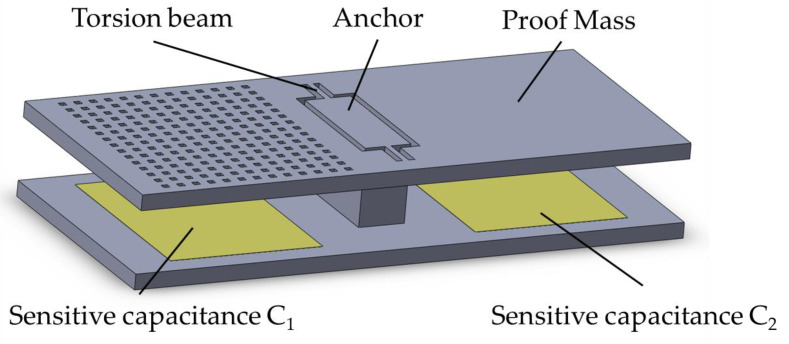
Pendulous capacitive micro-accelerometer.

**Figure 2 micromachines-14-00190-f002:**
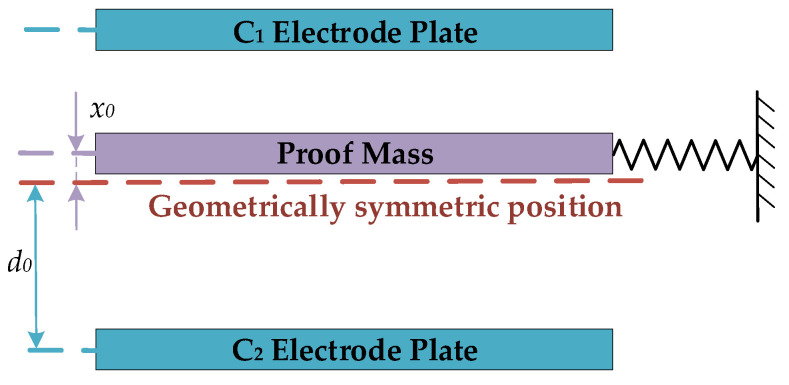
Simplified model of the capacitive micro-accelerometer.

**Figure 3 micromachines-14-00190-f003:**
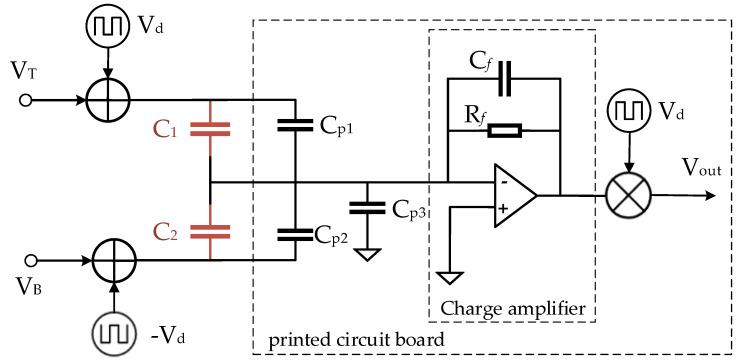
The electrical model of the measurement circuit.

**Figure 4 micromachines-14-00190-f004:**
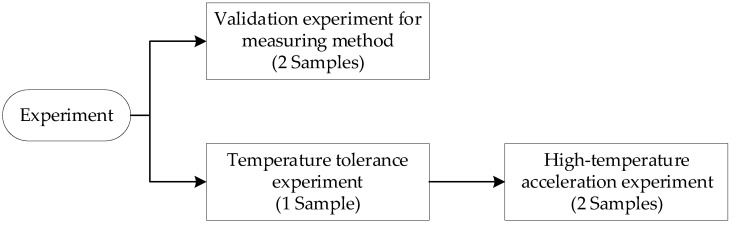
Experimental process.

**Figure 5 micromachines-14-00190-f005:**
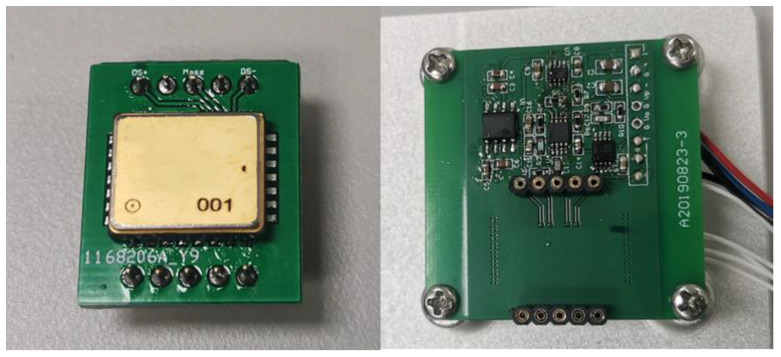
Experimental sample and measurement board.

**Figure 6 micromachines-14-00190-f006:**
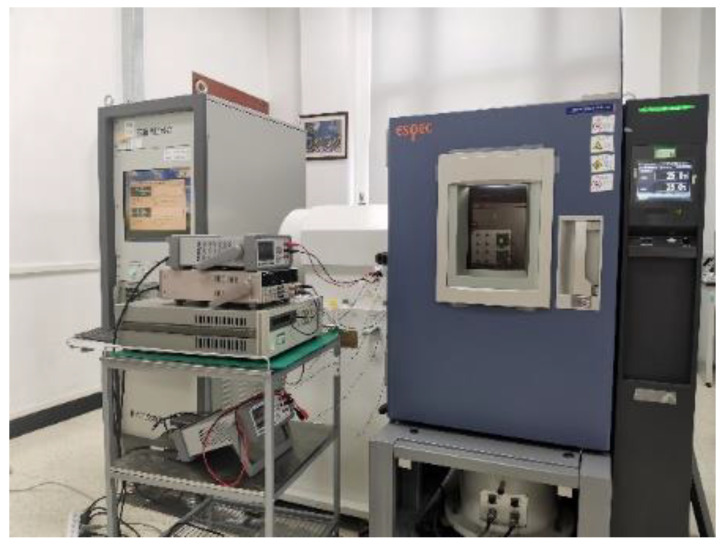
The measuring system.

**Figure 7 micromachines-14-00190-f007:**
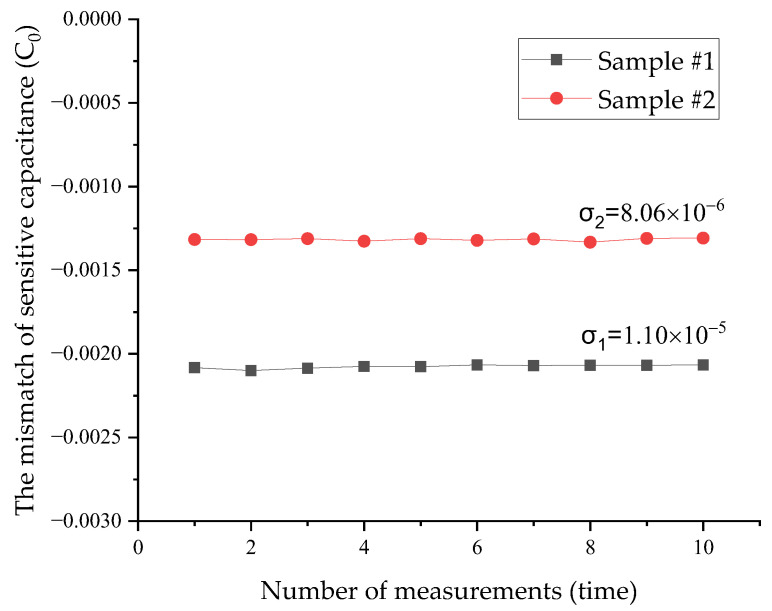
Measurement data of experimental samples.

**Figure 8 micromachines-14-00190-f008:**
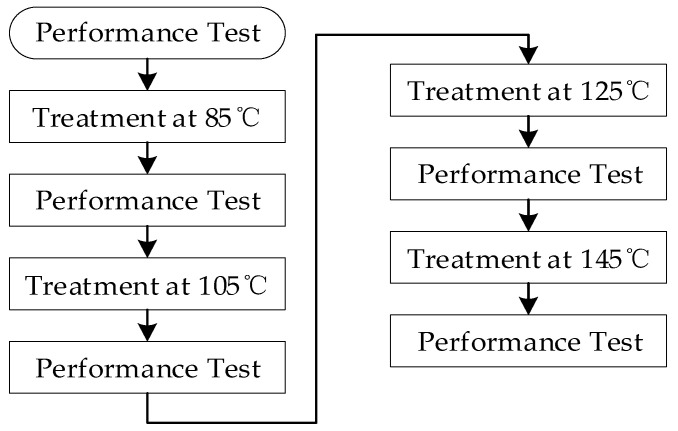
The process of the temperature tolerance experiment.

**Figure 9 micromachines-14-00190-f009:**
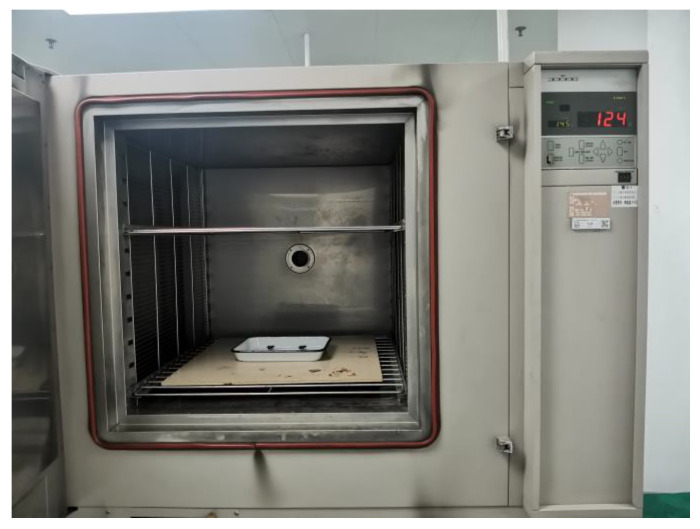
High-temperature accelerated test chamber.

**Figure 10 micromachines-14-00190-f010:**
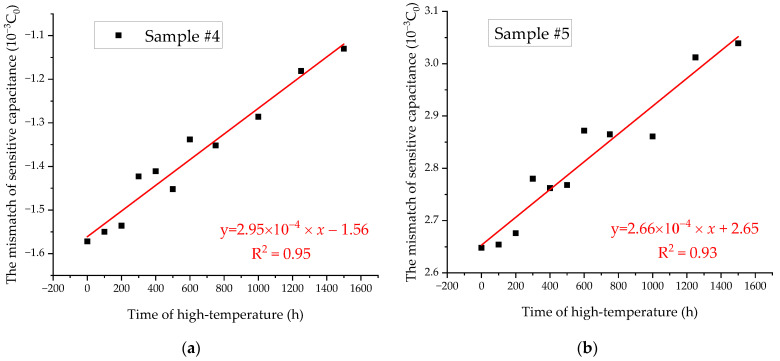
Sensitive capacitance mismatch of the high-temperature acceleration experiment: (**a**) Data of Sample #4; (**b**) Data of Sample #5.

**Figure 11 micromachines-14-00190-f011:**
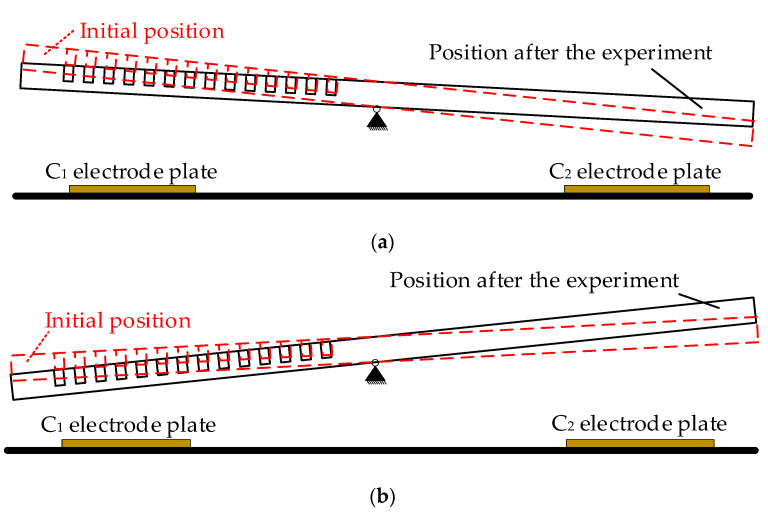
Schematic diagram of the position of the proof mass: (**a**) Sample #4; (**b**) Sample #5.

**Table 1 micromachines-14-00190-t001:** The characteristic parameters of the experimental sample.

Symbol	Description	Value	Unit
S	Electrode plate overlap area	1.56	mm^2^
d_0_	Average distance between electrode plates	1.5	um
C_0_	Sensitive capacitor	11	pF
Sen	Sensitivity of sensitive capacitor	96.35	fF/g

**Table 2 micromachines-14-00190-t002:** Measurement data of sample #1.

Measurement Point	*V_T_*	*V_B_*	*V* _Out_
A	0.00000	0.00000	0.17576
B	5.00870	5.00035	0.17971
C	5.01192	5.00035	0.17360

**Table 3 micromachines-14-00190-t003:** The electrode plate voltage of the sample at electrostatic force balance (unit: V).

NO.	Sample #1			Sample #2		
	*V_T_*	*V_B_*	*V_T_/V_B_*	*V_T_*	*V_B_*	*V_T_/V_B_*
1	5.01078	5.00035	1.00209	5.00695	5.00036	1.00132
2	5.01086	5.00035	1.00210	5.00694	5.00035	1.00132
3	5.01079	5.00035	1.00209	5.00691	5.00035	1.00131
4	5.01074	5.00035	1.00208	5.00699	5.00035	1.00133
5	5.01075	5.00035	1.00208	5.00691	5.00035	1.00131
6	5.01069	5.00035	1.00207	5.00696	5.00035	1.00132
7	5.01071	5.00035	1.00207	5.00691	5.00034	1.00131
8	5.01070	5.00035	1.00207	5.00701	5.00034	1.00133
9	5.01071	5.00035	1.00207	5.00689	5.00034	1.00131
10	5.01069	5.00035	1.00207	5.00689	5.00034	1.00131
STD of *V_T_/V_B_*	1.10 × 10^–5^			8.07 × 10^–6^		

**Table 4 micromachines-14-00190-t004:** The scale factor of the samples (unit: mV/g).

Measuring Point	Sample #3
Initial	338.12
85 °C	338.37
105 °C	338.26
125 °C	338.15
145 °C	338.07
Maximum change ratio	0.07%

**Table 5 micromachines-14-00190-t005:** Measurement data of high-temperature acceleration experiments (unit: 10^−3^
*C*_0_).

Measuring Point	Sample #4	Sample #5
Initial	−1.572	2.648
100 h	−1.550	2.654
200 h	−1.536	2.676
300 h	−1.423	2.780
400 h	−1.411	2.762
500 h	−1.452	2.768
600 h	−1.338	2.872
750 h	−1.352	2.865
1000 h	−1.286	2.861
1250 h	−1.181	3.012
1500 h	−1.130	3.039
Maximum change	+0.442	+0.391

## Data Availability

Not applicable.

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
