# Peer review of "Long-Term Degradation Evaluation of the Mismatch of Sensitive Capacitance in MEMS Accelerometers"

_micromachines, 2023, doi:10.3390/mi14010190_

Round 1

Reviewer 1 Report

The analysis and experiment results of the paper seems a lot odd, The principle described in formula seems have no relation with the experiment results, and the analysis method should be well introduced, The References seems not enough to support the analysis, if there any similar papers can be cited to verify the paper?

There were also a lot need to be described more detail listed below.

1 Formula 1 describe that the relationship between capacitance mismatch and plate distance, but K1-K4 seems help less in evaluating degradation of the mismatch of sensitive capacitance

2 The charge introduced by Vt and Vb is ignored in Formula 2 , Why.

3 Why Fe=0 in formula 3, why the stiffness of the accelerometer is ignored when the output voltage is U0 under the action of Vt and Vb .

4 Why Vt and Vb is set around 5V, but not 0V in Table 1

5 The capacitance error is 0.36fF, it seems a big value, and if the error is helpful for capacitance mismatch test, for the acceleration capacitance accuracy can be several aF now

6 is there any evidence to point that the decrease of residual stress is the main reason for the variation of sensitive capacitance mismatch in the paragraph under table 3

7 Figure 11 pointed that the position of mass changed after experiment, why the assumption is valid, and why the average distance between mass and capacitor plate is unchanged.  

8 Figure 10 shows the mass position degrades linearly, is there any similar report, it seems not valid.

Reviewer 2 Report

The authors investigated the influence of sensitive capacitance mismatch on the performance of MEMS accelerometer. In addition, they reported a method to measure the mismatch of sensitive capacitance. Experimental results indicated that the mismatch of sensitive capacitance varies linearly with time, and the variation rate is similar for the same batch of specimens. This manuscript can be enhanced based on the following issues:
1.- The abstract should improve the description of the main results. In addition, this section should add a conclusion.
2.- The introduction should incorporate more discussions on the main limitations of the methods reported in the literature for measuring the mismatch of sensitive capacitance in MEMS accelerometers.
3.-The introduction should include the main advantages and limitations of the proposed method compared to others reported in the literature.
4.- The second section should enhance the description of the proposed method. Which are the assumptions used in the proposed method?
5.-The authors should add more information on the main technical parameters of the MEMS accelerometers used in the experiments.
6.- The discussions of the experimental results should be improved.
7.- This manuscript could incorporate more discussions of the compensation circuits to decrease the adverse effect due to variations in sensitive capacitor mismatch.
8.- What is the future research work?

Round 2

Reviewer 2 Report

This manuscript version was enhanced by considering the reviewer's comments.